# The Role of CREBBP/EP300 and Its Therapeutic Implications in Hematological Malignancies

**DOI:** 10.3390/cancers15041219

**Published:** 2023-02-14

**Authors:** Yu Zhu, Zi Wang, Yanan Li, Hongling Peng, Jing Liu, Ji Zhang, Xiaojuan Xiao

**Affiliations:** 1Department of Hematology, The Second Xiangya Hospital, Molecular Biology Research Center, School of Life Sciences, Hunan Province Key Laboratory of Basic and Applied Hematology, Central South University, Changsha 410011, China; 2The Affiliated Nanhua Hospital, Department of Clinical Laboratory, Hengyang Medical School, University of South China, Hengyang 421001, China

**Keywords:** CREBBP, EP300, acetylation, hematological malignancy, chemoresistance, immunotherapy

## Abstract

**Simple Summary:**

Through their regulatory effects on gene expression, histone acetyltransferases have been implicated in the normal physiological activities and genesis of cancer. Genetic aberrations of *CREBBP/EP300* have been observed in various types of solid tumors and hematologic malignancies, making them serve as promising therapeutic targets. Here, this review discusses the critical role of CREBBP/EP300 in normal hematopoiesis and also provides a comprehensive overview of how they contribute to the genesis and progression of hematologic malignancies. The impact of different CREBBP/EP300 inhibitors and histone deacetylase inhibitors on targeting therapeutic potential, alleviating chemotherapy resistance, and enhancing immunotherapeutic potential has also been reviewed.

**Abstract:**

Disordered histone acetylation has emerged as a key mechanism in promoting hematological malignancies. CREB-binding protein (CREBBP) and E1A-binding protein P300 (EP300) are two key acetyltransferases and transcriptional cofactors that regulate gene expression by regulating the acetylation levels of histone proteins and non-histone proteins. *CREBBP/EP300* dysregulation and CREBBP/EP300-containing complexes are critical for the initiation, progression, and chemoresistance of hematological malignancies. CREBBP/EP300 also participate in tumor immune responses by regulating the differentiation and function of multiple immune cells. Currently, CREBBP/EP300 are attractive targets for drug development and are increasingly used as favorable tools in preclinical studies of hematological malignancies. In this review, we summarize the role of CREBBP/EP300 in normal hematopoiesis and highlight the pathogenic mechanisms of CREBBP/EP300 in hematological malignancies. Moreover, the research basis and potential future therapeutic implications of related inhibitors were also discussed from several aspects. This review represents an in-depth insight into the physiological and pathological significance of CREBBP/EP300 in hematology.

## 1. Introduction

Normal gene expression is indispensable for cells to perform hematopoiesis and other various physiological activities. Histone acetyltransferases (HATs) are a group of epigenetic modifying enzymes that play vital roles in the regulation of gene transcription. Among the HAT family, the acetyltransferase CREB-binding protein (CREBBP; also known as CBP or KAT2A) and E1A-binding protein P300 (EP300; also known as P300 or KAT2B) are two closely related regulators. Both are widely expressed within and outside the hematopoietic system and serve as tumor activators or suppressors depending on the situation [1,2]. CREBBP/EP300 regulate various key physiological functions, including cell apoptosis [3], proliferation and differentiation [4,5], DNA repair [6,7], and somatic cell reprogramming [8]. CREBBP/EP300 are essential for the maintenance of normal hematopoiesis. However, a growing number of studies have shown that aberrant expression of *CREBBP/EP300* is associated with tumorigenesis and the progression of hematological malignancies. Recently, abundant CREBBP/EP300 inhibitors have been developed, representing emerging tools for clinical intervention in human diseases [9,10]. Here, after providing a brief overview of the molecular characteristics and functions of CREBBP/EP300, we summarize the current knowledge about the implications of CREBBP/EP300 in normal hematopoiesis and hematological malignancies. Finally, we discuss some prospective therapeutic strategies for CREBBP/EP300 inhibitors in hematological malignancies.

### 1.1. Molecular Domains and Characteristics of CREBBP/EP300

CREBBP and paralogous EP300 are two evolutionarily conserved and functionally related enzymes, and both share a similar domain architecture consisting of a diverse set of protein- and DNA-binding modules [10] (Figure 1). The histone acetyltransferase (HAT) domain of CREBBP/EP300 interacts with histone and is the site of acyl-CoA binding, which provides the basis for the acetylation state of multiple interacting proteins. A variety of short-chain acyl-CoA variants can serve as substrates for EP300, such as propionyl-CoA, crotonyl-CoA, and butyryl-CoA. The X-ray crystal structure of EP300 bound to these acyl-CoAs has been determined, suggesting that the active site of EP300 could accommodate longer acyl chains without major structural rearrangement [11]. Furthermore, CREBBP/EP300 and histone deacetylases (HDACs) antagonistically regulate mRNA or protein stability through acetylation or deacetylation. For example, CREBBP/EP300 acetylate the exoribonuclease CAF1, a catalytic subunit of the CCR4-CAF1-NOT deadenylase complex, thereby promoting deadenylation and degradation of poly(A) RNA, while HDAC-mediated deacetylation enhances mRNA stability [12].

The bromodomain (BRD) of CREBBP/EP300 facilitates histone acetylation independent of chromatin localization. CREBBP/EP300 BRD inhibition leads to an enhancer-bias reduction in CDK9 at chromatin, indicating impaired positive transcription elongation factor (p-TEFb) complex recruitment and a concomitant reduction in RNA polymerase II at enhancers [13]. High-resolution X-ray crystal structures of the BRD-PHD tandem module of human CREBBP in complex with a lysine-acetylated histone H4 peptide have been solved at 1.9Å and 1.83Å resolution, respectively [14]. Furthermore, the autoacetylation of CREBBP is pivotal for the interaction of its BRD with H3K56ac on free histones with higher affinity than other monoacetylated binding partners. The interaction between CREBBP BRD and the histone chaperone anti-silencing function 1 (ASF1) is important for H3K56ac [15]. In diffuse large B-cell lymphoma (DLBCL), *CREBBP/EP300* mutations have been found to primarily affect the HAT-lysine acetyltransferase 11 (KAT11) domain [16,17]. The cell cycle regulatory domain 1 (CRD1) domain of EP300 mediates transcriptional repression. The ZZ-type zinc finger (ZZ) domains of CREBBP and EP300 have different activities leading to different biological functions. The ZZ domain of EP300 has been identified as a novel member of the family of epigenetic readers, which selects the histone H3 tail to promote H3K27ac and H3K18ac [18]. Moreover, destruction of the CREBBP really interesting new gene (RING) domain is a recurrent pathogenic event caused by various types of somatic alterations in acute lymphocytic leukemia (ALL) [19]. It has been reported that the IRF3 and STAT1 dimerization enable trans-autoacetylation of EP300 in a highly conserved and intrinsically disordered autoinhibitory lysine-rich loop (AIL), further causing HAT activation. Ortega et al. also found that substrate access to the active site of HAT involved a rearrangement of an autoinhibitory RING domain [20]. The crystal structure of the ternary complex formed by the kinase-inducible domain of CREB-interacting (KIX) domain of CREBBP/EP300, the activation domain of human T-cell leukemia virus I basic leucine zipper protein (HTLV-1 HBZ), and the activation domain of c-Myb has been reported. This complex has been associated with hematopoietic differentiation and the development of adult T-cell leukemia [21].

### 1.2. Transcriptional Regulatory Activity of CREBBP/EP300

CREBBP/EP300 can act as transcriptional co-activators participating in the transcriptional regulation of multiple proteins [22,23,24]. Genes interacting with CREBBP/EP300 include the Krüppel-like factor family, the *Gata* family, and the *Runx* family, which have important functions in hematopoiesis, cell cycle regulation, and self-renewal of hematopoietic stem cells (HSCs) [24,25,26]. *Ep300* deletion results in altered activity of specific enhancers in *Tet2*-deficient hematopoietic stem and progenitor cells (HSPCs) showing enrichment for key hematopoietic regulators, including *Ets*, *Runx*, *Gata*, *Irf8*, *NF-κB*, and *Myb* family members [27]. Inhibition of CREBBP/EP300 BRD reduces H3K27ac levels specifically at enhancers, and acute loss of H3K27ac from enhancers leads to suppression of nascent enhancer RNA (eRNA) production and significant downregulation of enhancer-driven oncogenes expressions, such as *MYC*, *MYB*, and *IRF4*, in several hematological cancer cells [13]. It leads to profound anti-proliferative effects in vitro and in vivo.

## 2. The Role of CREBBP/EP300 in Normal Hematopoiesis

CREBBP/EP300 act as tumor suppressors and maintain normal hematopoietic function. The *CREBBP* BRD mutation disrupts its interaction with ASF1 causing Rubinstein-Taybi syndrome, and patients are prone to develop cancer [15]. A full complement of functional Crebbp/Ep300 is required to maintain normal hematopoiesis, and mice with monoallelic inactivation of the *Crebbp* gene generate highly permeable multilineage defects in definitive hematopoietic differentiation and an increased incidence of hematological malignancies with age [28]. CREBBP/EP300 play different roles in the self-renewal of HSCs, with a full dose of CREBBP being essential for self-renewal and differentiation of HSCs, whereas EP300 is crucial for proper hematopoietic differentiation [25,29,30]. The interaction between CREBBP and c-Myb regulates HSCs and HSPCs function and is also critical for human acute myeloid leukemia (AML) [27,31,32]. The loss of *Ep300EP300* upregulates *Myb*, and *Myb* depletion inhibits the proliferation of HSPCs and improves the survival of leukemia-bearing mice [27]. The activity of at least one acetyltransferase of CREBBP and EP300 is indispensable for B-cell development and survival [22]. CREBBP and EP300 regulate common as well as distinct transcriptional targets in sub-compartments of the germinal center (GC), a stage of the humoral immune response where B-cells undergo immunoglobulin affinity maturation. It has been reported that *CREBBP* loss occurs at different stages of lymphopoiesis resulting in different functions. Specifically, mice with the early loss of *Crebbp* in the HSPCs compartment are more likely to develop aggressive lymphomas. In contrast, loss of *Crebbp* in committed lymphoid cells inhibited tumor progression. Furthermore, Horton et al. identified *CREBBP* mutation in HSPCs of *CREBBP*-mutant lymphoma patients, which has profound implications for the underlying cellular origin and subsequent evolution of lymphoid malignancies [33].

CREBBP/EP300 are associated with HSC maintenance. EP300 and MED12 overlap in 80% of enhancers and >70% of super-enhancers, and both robustly interact on the chromatin of mouse isolated HSPCs and HPC-7 cells to maintain the active state of hematopoietic enhancers. Deletion of MED12 destabilizes EP300 binding at lineage-specific enhancers, resulting in H3K27ac depletion, enhancer inactivation, and consequent loss of HSCs stemness signatures, such as *ERG*, *ETV6*, *RUNX1*, and *TCF7* [34]. CREBBP/EP300 are associated with hematopoietic microenvironment homeostasis. *Myb/Ep300* deficiencies induce megakaryocytosis and thrombocytosis, which further leads to a decrease in circulating thrombopoietin concentrations and significant perturbations in the HSCs compartment. *Crebbp* and *Brca1* functionally interact to maintain normal hematopoiesis. *Crebbp* haploinsufficiency in mice is deleterious for the bone marrow microenvironment, which results in myeloproliferation associated with an increase in splenic HSCs as well as a lethal systemic inflammatory disorder [35,36].

## 3. The Role of CREBBP/EP300 in Hematological Malignancies

### 3.1. Lymphoma

*CREBBP/EP300* gene mutations frequently occur in hematological malignancies through a variety of different mechanisms, resulting in poor patient prognosis, including worse overall survival (OS), progression-free survival (PFS), and event-free survival (EFS) [37,38,39,40,41]. Genetic abnormalities of *CREBBP/EP300* are frequent in hematological malignancies and include gene mutations, copy number variations, and structural variations (Table 1 and Table 2). For example, *CREBBP*/*EP300* gene mutations are frequently detected in diverse lymphomas, such as follicular lymphoma (FL) [39,40], DLBCL [16,42,43], in situ follicular neoplasia [44], peripheral T-cell lymphoma (PTCL) [37,45], angioimmunoblastic T-cell lymphoma (AITL) [37], and plasmablastic lymphoma (PBL) [46]. The frequency and type of *CREBBP/EP300* mutations vary significantly among lymphomas due to different geographic regions or subtypes. Firstly, *EP300* mutations are the most common type of epigenetic mutation in North American Adult T-cell leukemia–lymphoma (ATLL) (20%), approximately three times more than in Japanese ATLL [41]. Secondly, *EP300* mutations are more significant in AITL than in peripheral T-cell lymphoma–not otherwise specified (PTCL-NOS) (22.2% vs. 2.9%) [45]. Finally, Garcia-Ramirez et al. found a significant difference in *CREBBP* mutations between FL and DLBCL. *CREBBP* frameshift/nonsense mutations occurred more frequently in DLBCL compared to FL, but missense mutations were more frequently observed in FL. In addition, *CREBBP* mutations were associated with increased MYC expression in primary DLBCL tumors but not in FL [47]. Compared with *EP300*, mutational inactivation of *CREBBP* has additional cell-intrinsic engraftment and growth-promoting effects in orthotopic xenograft models of DLBCL [48]. Deletion of *Ep300*, but not *Crebbp*, impairs the fitness of GC B-cells in vivo. However, joint loss of *Crebbp* and *Ep300* completely abolished GC formation, suggesting EP300-dependency in *Crebbp*-deficient lymphoma cells [49,50]. 

*CREBBP/EP300* mutations as a major pathogenetic mechanism shared by common forms of B-cell non-Hodgkin lymphoma (B-NHL) have direct implications for the use of drugs targeting the acetylation/deacetylation mechanism. Pasqualucci et al. found that lymphomagenesis was related to inactivation or dose reduction of the HAT domain caused by the *CREBBP/EP300* mutation [51,52]. CREBBP/EP300 and BCL6/SMRT/HDAC3 exert opposing functions by regulating enhancer/super-enhancer networks with key roles in the GC reaction, including signal transduction from the B-cell receptor and CD40 receptor, NF-κB and Galpha13 signaling, T-cell-mediated B-cell activation, plasma cell differentiation, and major histocompatibility complex class II (MHCII) antigen processing and presentation [53,54]. Genes disturbed by *CREBBP* mutation are direct targets of the BCL6/SMRT/HDAC3 tumor–repressor complex. *CREBBP* loss of function contributes to lymphomagenesis by promoting immune escape in vitro and in vivo [42,55,56] (Figure 2). Histone deacetylase inhibitors (HDACis) play positive roles in the treatment of patients with FL, especially those with nonfunctional CREBBP [57]. These findings suggest that HDACis are available for treating B-NHL patients, as they may contribute to re-establishing physiologic acetylation levels and subsequently contribute to the restoration of tumor immune surveillance. Certainly, efficacy and target specificity should be evaluated.

### 3.2. Leukemia

Genetic abnormalities of *CREBBP/EP300* are also common in leukemia. For example, *CREBBP/EP300* mutations frequently occur in primary and relapsed pediatric ALL [58,59,60,61,62,63,64], aggressive NK-cell leukemia (ANKL) [65], and chronic-phase chronic myeloid leukemia (CML-CP) [66]. *CREBBP* mutation is associated with hyperdiploid karyotype and *KRAS* mutation in relapsed pediatric ALL. *CREBBP* knockdown enhances RAS/RAF/MEK/ERK signaling in ALL with Ras pathway mutation but remains sensitive to MEK inhibitors [67]. An analysis found that 18.3% of relapse ALL cases (n = 71) had sequence or deletion mutations of *CREBBP*, which resulted in truncated alleles or detrimental substitutions in conserved residues of the HAT domain [68]. In *MLL1*-rearranged acute leukemia, EZH2 binds directly to cMyc and EP300 via a hidden transactivation domain, thereby mediating gene activation and contributing to tumorigenesis [69]. CN470 has notable anti-tumor activity against *MLL1*-rearranged ALL by binding to the bromodomains (BRDs) of BRD4, CREBBP, and EP300 in vitro and in vivo [70].

CREBBP/EP300 play an important role in leukemogenesis by participating in the regulation of various fusion proteins and key proteins (Table 3). The gene expression profile of *TCF3-ZNF384*-positive ALL is related to *CREBBP-ZNF384*- and *EP300-ZNF384*-positive ALL, but not to other conventional genetic subtypes [71,72,73,74]. *MOZ* (*MYST3*)–*CREBBP*, t(8;16) (p11;p13), is a very rare abnormality in AML [75,76]. RUNX1 (AML1) is a common target of chromosomal translocations and has been suggested to be directly acetylated by EP300 at residue K43 in several types of AML and ALL [77,78]. MOZ regulates gene transcription by activating the RUNX1 transcription factor complex and is involved in the regulation of AML development and erythrophagocytosis [76,79,80].

EP300 is abundant at super-enhancers and coincident with sites of GATA1 and MYC occupancy in chronic myeloid leukemia (CML) cell line K562. BRD inhibitors interfere with these oncogene-driven transcriptional programs, leading to cell cycle arrest in the G0/G1 phase [5] (Figure 3A). In AML, 21q22/*HMGN*1 amplification cooperates with the AML-ETO9a to impair myeloid differentiation and enhance leukemia stem cell activity. HMGN1 overexpression increases chromatin accessibility, expression, and H3K27ac at loci important for HSCs and leukemia, including many CREBBP/EP300 targets [93] (Figure 3B). Although the above data show the tumor-promoting role of CREBBP/EP300, they sometimes exert tumor-suppressive functions as well. For example, *Ep300*, but not *Crebbp*, plays a critical role in blocking the transformation of myelodysplastic syndrome (MDS) to AML and in controlling the balance between symmetric stem cell self-renewing divisions and stem cell depleting divisions in Nup98-HoxD13 transgenic mice, an animal model that phenotypically replicates human MDS [94] (Figure 3C). Promoting the senescence of malignant tumor cells is also a therapeutic strategy. Deletion of *β-Arrestin1* promotes senescence of B-ALL initiating cells in vitro and in vivo by inhibiting the EP300/Sp1 interaction at −28 to −36 bp of the *hTERT* promoter [95,96] (Figure 3D).

### 3.3. Multiple Myeloma

In a next-generation sequencing analysis of MM, *EP300* (11.6%, n = 147) was one of the most commonly mutated genes in chromatin regulators [97]. CREBBP/EP300 exhibit evolutionary conservation in maintaining the core transcription program that is dynamically repressed following acute lysine acetyltransferase inhibition. Acute catalytic CREBBP/EP300 inhibition modulates transcription independently of DNA accessibility and selectively suppresses transcription of distinct oncogenic networks in different cancer types. Robust CREBBP/EP300 activity is required to maintain genome-wide histone acetylation and facilitates the recruitment of co-activators and RNA Pol II in the face of rapid deacetylation kinetics. HDAC3-dependent NCoR/SMRT co-repressor complexes and CREBBP/EP300 co-locate in active regions of chromatin, which are functionally antagonistic. Compensatory loss of NCoR/SMRT complexes perturbs genome-wide histone deacetylation rates and mitigates the transcriptional effects of CREBBP/EP300 inhibition. However, these effects remain highly reversible prior to compensatory histone methylation switching, and deacetylation of H3K27 provides nucleation sites for reciprocal methylation switching, a feature that can be treated by concomitant repression of KDM6A/UTX and CREBBP/EP300 in MM [54] (Figure 4). Shed syndecan-1 is a cell surface heparan sulfate proteoglycan that is translocated to the nucleus and binds to EP300 to inhibit histone acetylation in myeloma cells and bone-marrow-derived stromal cells, thereby facilitating communication within the tumor microenvironment [98].

The IRF4/MYC axis is critical for MM progression. Inhibition of CREBBP/EP300 BRDs leads to a significant reduction in H3K18ac and H3K27ac at the *IRF4* super-enhancer and transcription start site, which directly inhibits *IRF4* and its downstream target genes such as *cMYC*, resulting in decreased viability of MM cell lines and causing cell cycle arrest and apoptosis (Figure 4). The ectopic expression of IRF4 and MYC antagonizes the phenotypic and transcriptional effects of CREBBP/EP300 BRD inhibition [99]. In MM, cereblon has been identified as a primary target of immunomodulatory drugs (IMiDs), such as lenalidomide [100]. Recently, IL6 upregulation and activation of STAT3 and its downstream genes (*PIM2* and *BIRC5*) have been identified as a critical resistance mechanism to IMiDs. BRD inhibitors re-sensitize several IMiD-resistant human MM cell lines to lenalidomide by targeting the IRF4/MYC axis and also increase the sensitivity of IMiD-sensitive cell lines to lenalidomide. SGC-CBP30 and lenalidomide have a synergistic effect in reducing myeloma cell viability [101]. EP300/Sp1 interaction is also present in MM, and this complex promotes cell proliferation by regulating *IQGAP1* transcription at the promoter [102].

### 3.4. Myelodysplastic Syndromes

MDS is a clonal bone marrow stem cell disorder and a third of patients develop AML. A large-scale genomics study found that the frequency of *CREBBP/EP300* mutations in MDS patients is about 7% [103]. Several studies have shown that *CREBBP* is one of the genes affected by chromosomal translocations in patients with therapy-related MDS [24]. Previously, Kojima et al. reported a case of therapy-related MDS with chromosome t(10;16)(q22;p13), in which a *MORF*-*CREBBP* fusion variant was detected [104]. In addition, some MDS patients have t(11;16)(q23;p13) in which the *MLL* gene is fused to the *CREBBP* gene [105,106,107]. *Crebbp^+/−^* mice consistently develop MDS/myeloproliferative neoplasms (MPN) at 9–12 months of age and are hypersensitive to γ-radiation. Meanwhile, mice exhibit reduced numbers of HSCs and common myeloid progenitors and increased granulocyte/macrophage progenitors [108]. Interestingly, EP300 also has tumor suppressor effects in several different clinically relevant MDS models driven by mutations in epigenetic regulators TET2, ASXL1, and SRSF2, respectively, inhibiting the malignant transition of MDS to AML [27,94]. This suggests a potential therapeutic application of EP300 agonists in the treatment of MDS with the aforementioned inactivating mutations.

## 4. The Therapeutic Implications of CREBBP/EP300

### 4.1. Application of CREBBP/EP300 Small Molecule Compounds

#### 4.1.1. CREBBP/EP300 Agonists

Aberrant expression of *CREBBP/EP300* causes Rubinstein–Taybi syndrome and multiple hematologic malignancies by losing their activity [15,81,82]. *CREBBP* is a haploinsufficient tumor suppressor gene in GC B-cells. *CREBBP* deficiency promotes the development of B-cell lymphoma [42,47,48,53]. EP300 could suppress the transition of MDS to AML, suggesting the therapeutic potential of EP300 agonists in MDS patients with *Tet2*-inactivated mutations [27,94]. YF-2 is a highly selective CREBBP/EP300 HAT domain agonist, and it could improve the genome editing efficiency of cas9 nucleases [109]. CTB, an agonist of EP300, increases the expression of EP300 to induce the acetylation of P53 protein and subsequently leads to the death of breast cancer cells. Therefore, CTB can be used as an anticancer drug [110]. TTK21 is also a CREBBP/EP300 agonist that conjugates glucose-based carbon nanosphere (CSP) to cross the blood–brain barrier, which is beneficial for adult neurogenesis and long-term memory function [111]. Currently, the number of CREBBP/EP300 agonists is relatively rare compared to inhibitors, and more in-depth studies are required.

#### 4.1.2. CREBBP/EP300 Inhibitors in Hematological Malignancies

Because of the BRD and acetyltransferase activity of CREBBP/EP300, HDACis, HAT domain inhibitors, and BRD inhibitors could inhibit their function. Researchers have summarized the development of CREBBP/EP300 inhibitors in recent years [9,10]. He et al. summarized the structures and molecular original names of 75 CREBBP/EP300 inhibitors from 2010 to 2021, including BRD inhibitors and HAT domain inhibitors, and they also discussed structure–activity relationship studies, bindings models, and biochemical data [10]. Xiong et al. discussed some candidates in clinical trials that could potentially inhibit CREBBP/EP300 function [9]. Chen et al. reviewed the structure and anticancer effects of small molecule compounds of CREBBP/EP300 [2]. In this review, we focus on summarizing several representative CREBBP/EP300 BRD inhibitors (CCS1477, CPI-637, and GNE272, etc.) and HAT inhibitors (A485, A-241, and C646, etc.) and their functions in hematological malignancies (Table 4).

To date, there is sufficient evidence that HDACis or other inhibitors of epigenetic modification molecules, alone or in combination, are effective against some hematologic malignancies with aberrant *CREBBP/EP300* expression [124]. Frequently mutated *EP300* in North American ATLL could be targeted by DNA methyltransferase inhibitor decitabine [41]. The phase II oral HDACi tucidinostat mitigates the negative prognostic impact of *CREBBP/EP300* mutations on DLBCL (ClinicalTrials.gov: NCT02753647) [125,126]. *CREBBP/EP300*-mutated DLBCL cells are dependent on CARM1 arginine methylation activity. Inhibition of CARM1 leads to H3K27ac deletion in the CREBBP/EP300 chromatin binding region, downregulating CREBBP target genes and further inhibiting DLBCL cell growth. The combination of CARM1 inhibitors and CREBBP/EP300 inhibitors may have significant therapeutic potential for the treatment of DLBCL, MM, and AML [127,128,129].

In addition to direct inhibition of CREBBP/EP300, blocking the interaction of CREBBP/EP300 with other oncogenic molecules is also an important anti-cancer pathway. XX-650-23, a small molecule inhibitor of CREB, specifically disrupts the interaction of CREB to the KIX domain of CREBBP and inhibits CREB-driven gene expression in AML cells, further promoting apoptosis and cell cycle arrest and prolonging survival in mice injected with human AML cells [130]. ICG-001 and C-82/PRI-724 specifically inhibit the interaction between CREBBP/β-catenin, but not EP300/β-catenin, thereby rendering drug-resistant CML-initiating cells to be sensitive to BCR-ABL tyrosine kinase inhibitors [131,132,133]. PRI-724 is currently in phase I clinical trials for AML and phase II clinical trials for CML (ClinicalTrials.gov: NCT01606579) [134]. MYBMIM, a peptidomimetic inhibitor, possesses significant anti-leukemia efficacy in vivo and in vitro. It blocks the assembly of the MYB: CREBBP/EP300 complex, further promoting apoptosis and inhibiting the growth and survival of AML cells. In addition, it has been suggested that MYBMIM affects promoters and enhancers of several transcription factors, including ERG, PU.1, CEBPA, and RUNX1, and suppresses gene expression, including *BCL2*, *MYC*, *GFI1*, *MTL5*, and *IKZF1* [135]. CRYBMIM, a second-generation version of MYBMIM, could specifically target the KIX domain of CREBBP/EP300 with a higher affinity for AML cells but less impact on normal hematopoietic progenitors [136]. Chetomin, a metabolite complex, could improve the poor prognosis of MM patients by targeting the HIF-1α/EP300 complex [137].

#### 4.1.3. CREBBP/EP300 Inhibitors in Clinical Trials

In addition to the clinical non-CREBBP/EP300 inhibitors mentioned above (tucidinostat and PRI-724), two targeted inhibitors of CREBBP/EP300 are currently in clinical trials in cancer patients. CCS1477, a CREBBP/EP300 BRD inhibitor developed by CellCentric, is in phase I/IIa clinical trials (ClinicalTrials.gov: NCT03568656) for the treatment of metastatic castration-resistant prostate cancer, metastatic breast cancer, and non-small cell lung cancer [138]. CCS1477 is also being evaluated in phase I/IIa clinical trials (ClinicalTrials.gov: NCT04068597) for the treatment of B-NHL, AML, MM, and higher-risk MDS [139]. FT-7051, a CREBBP/EP300 BRD inhibitor, has been tested for its safety and tolerability in patients with metastatic castration-resistant prostate cancer (ClinicalTrials.gov: NCT04575766) [140]. In summary, inhibitors of CREBBP/EP300 are gradually increasing and they are becoming more selective, powerful, and efficient, such as UMB298 [141], dCBP-1 [142,143], and B026 [121]. The current inhibitors could be useful tools to study CREBBP/EP300-related diseases. However, there are still few clinically available inhibitors. These data are useful for the development of novel and more effective CREBBP/EP300 inhibitors and the treatment of patients with related tumors.

### 4.2. CREBBP/EP300 and Chemoresistance in Hematological Malignancies

Transcriptional or epigenetic dysregulation of *CREBBP/EP300* in hematological malignancies may lead to chemoresistance. Zhou et al. found that CREBBP/EP300 inhibitors inhibited RTKs and the downstream activation of MAPK/ERK signaling, thereby overcoming mantle cell lymphoma (MCL) cells’ resistance to idelalisib in vitro and in vivo [144]. Downregulating CREBBP inhibits ALL cell proliferation and cell cycle progression and leads to daunorubicin resistance by interacting with E2F3a [145]. Furthermore, targeting CREBBP/EP300 restores the sensitivity of human MM cells to IMiDs [101].

Mutations in the histone modification gene may alter the function of proteins in the regulation of chromatin state, sensitizing tumor patients to HDACis, valproic acid, suberoylanilide hydroxamic acid, romidepsin, and chidamide. For example, PTCL patients bearing *CREBBP/EP300* mutations may respond to chidamide, either alone or in combination with other chemotherapy such as decitabine [45]. Moreover, *CREBBP* inactivation could promote the sensitivity of drug-resistant DLBCL cells to chidamide by regulating the cell cycle. The combination of AURKA inhibitor and chidamide is a novel therapeutic strategy for the treatment of relapsed/refractory DLBCL [146]. DLBCL cell lines expressing high BCL6 levels or *CREBBP/EP300* mutations are sensitive to GSK-J4, a histone demethylase KDM6B inhibitor [147]. A group of T-ALL cell lines with the mutated *CREBBP* allele were resistant to glucocorticoid dexamethasone treatment but sensitive to clinically useful concentrations of class I/II HDACi vorinostat [68]. However, Tamai et al. failed to find an association between the mutational status of the *CREBBP* gene and dexamethasone sensitivity in pre-B ALL. They suggested that most T-ALL cell lines were resistant to dexamethasone regardless of the mutational status of *CREBBP* [148]. Regardless of *CREBBP* mutation states and chromosomal aberrations, ICG-001 eradicated chemoresistance in vitro and significantly prolonged the survival of NOD/SCID mice engrafted with primary ALL [149]. Moreover, XX-650–23 and ICG-001 could synergize with dasatinib to enhance the sensitivity of pre-BCR^+^ ALL cells to dasatinib [150]. ICG-001 and HDACis have also been found to potentially relieve CML resistance to imatinib in vivo and in vitro [132,151]. In conclusion, patients with *CREBBP/EP300* mutations are sensitive to HDACis, and HDACis may help alleviate chemoresistance to other drugs. More drug combination strategies still need to be explored, and the mechanism of CREBBP/EP300 inhibitors leading to chemotherapy resistance is also worth further research.

### 4.3. Implications of CREBBP/EP300 in Immunotherapy

CREBBP/EP300 participate in tumor immune regulation by modulating the function of immune cells through various pathways. T regulatory cells (Tregs) are a subset of CD4^+^ T-cells with significant immunosuppressive effects. CREBBP/EP300 regulate the differentiation of Tregs through transcriptional and non-transcriptional mechanisms [114]. Tregs are downregulated in FL patients carrying *CREBBP/EP300* loss of function mutations. Inhibition of CREBBP/EP300 with GNE-781 impairs the differentiation of human CD4^+^ T-cells into Tregs, providing a therapeutic approach by promoting a pro-inflammatory tumor microenvironment [114]. In DLBCL, *CREBBP/EP300* mutations promote tumor-associated macrophage M2 polarization, thereby facilitating tumor progression in vivo and in vitro [16]. Trim24, a CREBBP-associated E3 ligase, promotes the recruitment of CREBBP to STAT6 by catalyzing the ubiquitination of CREBBP at lysine 119. However, the loss of *Trim24* promotes macrophage M2 polarization of mouse and human macrophages by inhibiting acetylation STAT6 at lysine 383, thus potentially suppressing anti-tumor immune responses [152]. The development of T helper 17 cells is regulated by the RORγt–SRC-3–EP300 axis, which is associated with T helper 17 cell-driven autoimmune diseases [153,154]. Tumor immune surveillance of NK cells is mediated by cytotoxicity receptor natural-killer group 2 member D (NKG2D), whose ligand NKG2D-ligand is not normally expressed in healthy cells. CREBBP/EP300 contribute to the upregulation of NKG2D ligands MICA/B and ULBP2 in humans and RAE-1 in mice. Inhibition of CREBBP/EP300 abrogates the sensitivity of stressed cells to NK-cell-mediated killing [22]. In MM, *IRF4* is the transcription target of *cMYC* and the repressor of *MICA* promoter activity. CREBBP/EP300 bromodomain and extra-terminal domain inhibitors (BETi) enhance NK-cell-mediated cytotoxicity by downregulating IRF4 and cMyc expression [117,155]. Finally, CREBBP/EP300 also regulate myeloid-derived suppressor cell (MDSC)-associated genes and their function by modulating H3K27ac of STAT-associated genes. BRD of CREBBP/EP300 may be targeted to boost anti-tumor immune response [156].

*CREBBP* mutations have been identified as an early event in FL evolution that is enriched within lymphoma cell progenitors, which reduces the expression of MHC II on tumor B-cells, thereby promoting immune evasion by reducing antigen presentation [157]. Mice that lack Crebbp exhibit hyperproliferation of GC B-cells upon immunization, which tends to cause myc-driven lymphomagenesis [48]. HDAC3 inhibitors reverse the abnormal epigenetic program caused by *CREBBP* mutations and recover immune surveillance by inducing the BCL6-suppressed IFN pathway and antigen-presenting genes. The synergistic effect of HDAC3 inhibitors and PD-L1 blockade restores the ability of tumor-infiltrating lymphocytes to kill DLBCL cells in a major histocompatibility complex class I (MHCI) and MHCII-dependent manner [56]. In addition, Xian et al. found that *CREBBP* and *CIITA* co-deletions/mutations were related to immune evasion [17,158]. Taken together, these results suggest that CREBBP/EP300 are involved in tumor immune responses by regulating immune cell functions, including Tregs differentiation, macrophage M2 polarization, and immune surveillance and killing functions of NK cells. *CREBBP/EP300* mutations lead to loss of MHCII expression in tumor cells and subsequently promote the development and progression of hematologic malignancies by facilitating immune escape. CERBBP/EP300 inhibitors and HDACis may help to restore and enhance the ability of immune cells to kill tumor cells.

## 5. Conclusions

As key acetyltransferase family members and transcriptional co-activators, CREBBP/EP300 regulate the acetylation levels of multiple substrates involved in the regulation of normal hematopoietic maintenance and the development of hematological malignancies. CREBBP/EP300 are involved in extensive hematopoietic regulatory networks at the transcriptome level [1]. The association between these two enzymes and other omics, such as metabolomics, still needs to be uncovered. Aberrant *CREBBP/EP300* expression is common in hematological malignancies and is associated with chemoresistance. These abnormal cells are sensitive to targeted inhibitors of CREBBP/EP300 and other epigenetic inhibitors, such as HDACis, DNA methyltransferase inhibitors, and histone demethylase inhibitors. Proteolysis-targeting chimeras (PROTACs) are small molecule compounds to induce the degradation of targeted proteins via ubiquitin–proteasome system [159]. JQAD1 and dCBP-1 are chemical degraders of CREBBP/EP300, and both have higher activity compared with HAT domain and BRD inhibitors [143,160]. dCBP-1 hijacks E3 ubiquitin ligase CRBN to selectively target CREBBP/EP300 for degradation. Degradation of CREBBP/EP300 kills MM cells by suppressing the oncogenic enhancer activity that drives *MYC* expression [143]. The efficacy and mechanism of JQAD1 and dCBP-1 in hematologic malignancies need further exploration. Currently, although more targeted and potential CREBBP/EP300 inhibitors are gradually increasing, most of them are in the preclinical experimental stage and clinically available inhibitors are still lacking. CREBBP/EP300 also function as tumor suppressors, but the number of agonists and relevant preclinical studies is relatively rare, which requires further research. A more in-depth study of the crystal structure and domain function of CREBBP/EP300 may facilitate the development of new inhibitors and agonists.

Because of the high mutability of *CREBBP/EP300*, genome-scale CRISPR-Cas9-based synthetic lethality screens can be used to discover the genes that exert the synthetic lethal effect, which may enhance the therapeutic potential of CREBBP/EP300. EP300 recruits BRD4 to the BATF promoter region, and targeting the BRD4-EP300 signaling cascade supports the generation of superior antitumor T-cell engraftment for adoptive immunotherapy [161]. FL patients have been found to harbor competent CD8^+^ T-cells specific for recurrent mutations, such as *CREBBP*^R1446C^. Production of off-the-shelf mutation-specific TCR vectors for T-cell engineering may be practical to further mitigate the adverse effects of CD19-CAR T-cell therapy by targeting tumor-specific mutations rather than the pan-B-cell marker CD19 [162]. It is instructive to design personalized immunotherapy for hematological malignancies with *CREBBP/EP300* driver mutations using next-generation sequencing technologies.

## Figures and Tables

**Figure 1 cancers-15-01219-f001:**
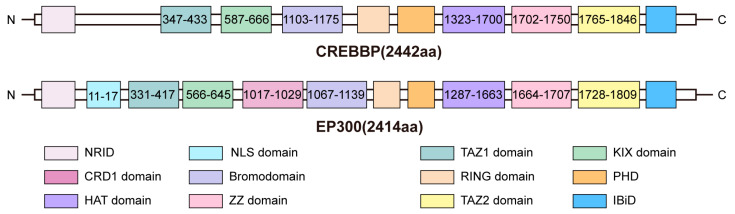
Molecular domains of the human CREBBP/EP300. CREBBP and EP300 have similar domain components and also share high sequence homology. Each domain is represented by a different color, and the number represents the location of the domain. NRID: nuclear receptor interaction domain; NLS: nuclear location signal; TAZ1: transcriptional-adaptor zinc-finger domain 1; KIX: kinase-inducible domain of CREB-interacting domain; CRD1: cell cycle regulatory domain 1; RING: really interesting new gene; PHD: plant homeodomain; HAT: histone acetyltransferase; ZZ: ZZ-type zinc finger; TAZ2: transcriptional-adaptor zinc-finger domain 2; IBiD: interferon-binding domain.

**Figure 2 cancers-15-01219-f002:**
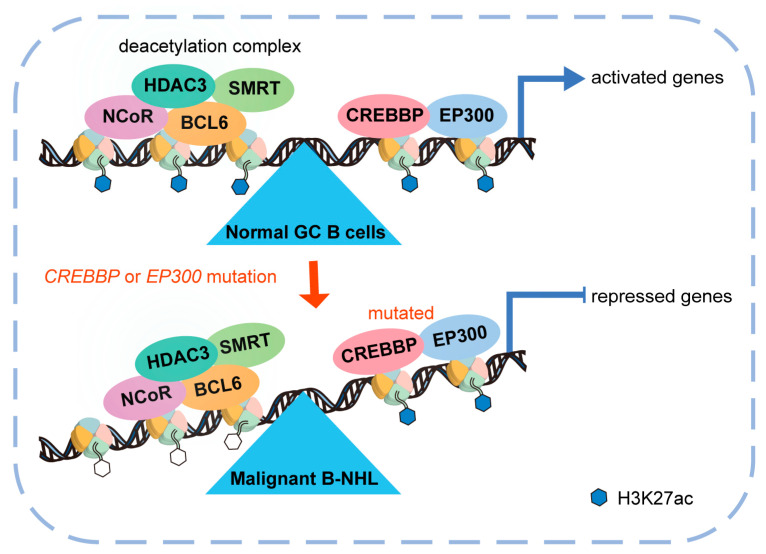
The opposite function between CREBBP/EP300 and the BCL6/SMRT/HDAC3 tumor–repressor complex in the germinal center reaction. In normal germinal center B-cells, CREBBP/EP300-mediated H3K27ac and BCL6/SMRT/HDAC3 complex-mediated deacetylation keep in balance. However, *CREBBP/EP300* mutation results in focal depletion of enhancer H3K27ac and aberrant transcriptional silencing of genes that regulate B-cell signaling and immune responses. It contributes to the development of HDAC3-dependent lymphomas by mediating immune escape. GC: germinal center; B-NHL: B-cell non-Hodgkin lymphoma.

**Figure 3 cancers-15-01219-f003:**
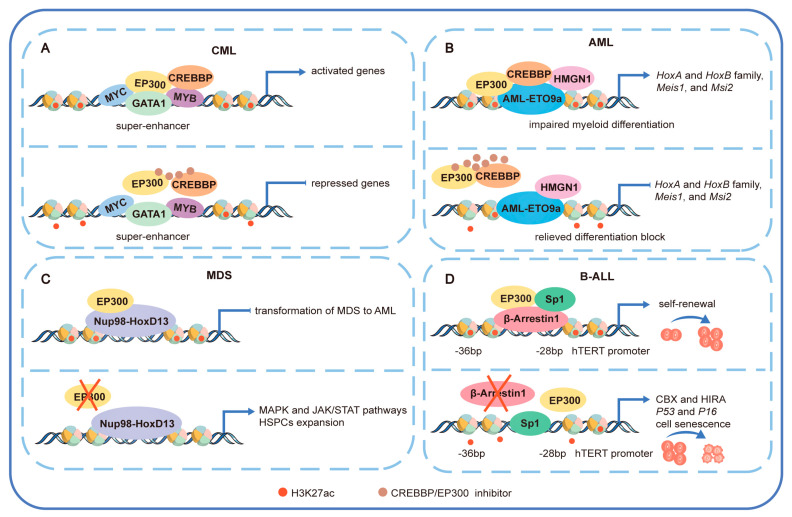
CREBBP/EP300 interact with multiple proteins to induce leukemogenesis. (**A**) In CML, CREBBP/EP300 are abundant in super-enhancer regions of oncogenes *GATA1*, *MYC*, and *MYB*. The CREBBP/EP300 BRD inhibitor reduces the levels of H3K27ac, inhibits the expression of *GATA1*, *MYC*, and their target genes, such as *TET1*, *FOSL1*, and *CCND1*, and causes cell cycle arrest. (**B**) HMGN1 overexpression cooperates with AML-ETO9a, which impairs myeloid differentiation and increases the expression of many CREBBP/EP300 targets, such as the *HoxA* and *HoxB* family, *Meis1*, and *Msi2*. CREBBP/EP300 inhibitors decrease H3K27ac in HMGN1-overexpression progenitors and relieve HMGN1-associated differentiation impairment. (**C**) EP300 plays a powerful role as a tumor suppressor in Nup98-HoxD13-driven progression of MDS to AML. Deletion of *EP300* promotes activation of the MAPK and JAK/STAT pathways in the HSPC compartment. Loss of *EP300* significantly triggers HSPCs’ expansion and accelerates MDS-associated leukemogenesis. (**D**) Depletion of *β-Arrestin1* reduced the interaction of EP300 with Sp1 at the *hTERT* promoter, which enhanced the expression of proteins (CBX and HIRA) and genes (*P53* and *P16*) associated with senescence, downregulated *hTERT* transcription, decreased telomerase activity, and shortened telomere length, thereby promoting senescence in leukemic cells. CML: chronic myeloid leukemia; AML: acute myeloid leukemia; MDS: myelodysplastic syndrome; B-ALL: B-cell acute lymphocytic leukemia.

**Figure 4 cancers-15-01219-f004:**
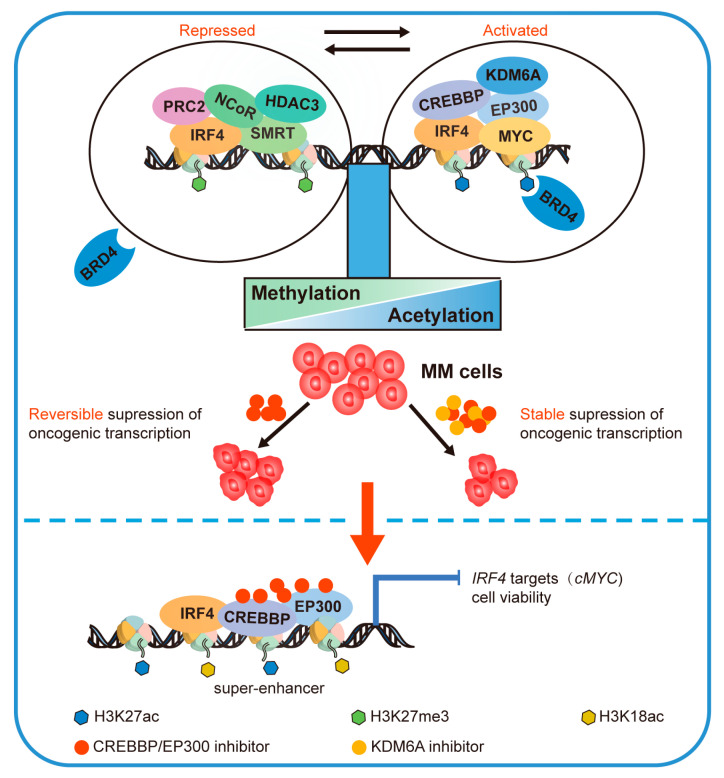
Oncogenic IRF4/MYC expression is regulated by histone acetylation and methylation switch. HDAC3-dependent NCoR/SMRT co-repressor complexes and PRC2-mediated methylation keep histone modifications in a hypoacetylated state and therefore oncogenes are repressed. However, CREBBP/EP300- and KDM6A-mediated demethylation activate oncogenes. The major effects of acute CREBBP/EP300 inhibition on acetylation and gene expression are highly reversible. The loss of H3K27ac and BRD4 binding represents transcriptional hypersensitivity to CREBBP/EP300 inhibition. CREBBP/EP300 inhibitors decrease H3K27ac and H3K18ac of the *IRF4* super-enhancer, inhibiting *IRF4* and its downstream target genes *cMYC*, further resulting in cell cycle arrest and apoptosis of MM cells. When combined with the CREBBP/EP300 inhibitor and the KDM6A inhibitor, the therapeutic effect was significantly improved. MM: multiple myeloma.

**Table 1 cancers-15-01219-t001:** *CREBBP/EP300* with high-frequency mutations in hematological malignancies.

Tumor	Gene	Mutation Number	Case Number with Mutation	Percentage (Total Number)
Acute Lymphoblastic Leukemia	*CREBBP*	2	1	1.1% (93)
Acute Myeloid Leukemia	*CREBBP*	4	4	2.0% (200)
	*EP300*	5	5	2.5% (200)
Chronic Lymphocytic Leukemia	*CREBBP*	2	2	1.3% (160)
	*EP300*	1	1	1.0% (105)
Cutaneous T-cell Lymphoma	*CREBBP*	2	2	4.7% (43)
Diffuse Large B-cell Lymphoma	*CREBBP*	35	28	20.7% (135)
	*EP300*	4	4	7.5% (53)
Hypodiploid Acute Lymphoid Leukemia	*CREBBP*	7	7	15.9% (44)
Lymphoma Cell Lines	*CREBBP*	10	9	26.5% (34)
	*EP300*	10	6	17.6% (34)
Lymphoid Neoplasm Diffuse Large B-cell Lymphoma	*CREBBP*	6	6	12.5% (48)
	*EP300*	3	3	3.6% (48)
Mature B-cell malignancies	*CREBBP*	229	180	23.8% (755)
	*EP300*	57	56	7.4% (755)
Multiple Myeloma	*CREBBP*	1	1	0.5% (205)
	*EP300*	1	1	0.5% (205)
Non-Hodgkin Lymphoma	*CREBBP*	3	2	14.3% (14)
	*EP300*	1	1	7.1% (14)
Pediatric Acute Lymphoid Leukemia	*CREBBP*	6	6	4.0% (150)
	*EP300*	2	2	1.3% (150)

Notes: Data mainly come from the TCGA database.

**Table 2 cancers-15-01219-t002:** *CREBBP/EP300* with abnormal copy numbers and structural variants in hematological malignancies.

Tumor	Gene	Type of CNA	Case Number with CNA	Percentage (Total Number)
Acute Myeloid Leukemia	*CREBBP*	AMP	1	0.5% (191)
Diffuse Large B-Cell Lymphoma	*CREBBP*	AMP	1	2.1% (48)
Lymphoid Neoplasm Diffuse Large B-cell Lymphoma	*CREBBP*	AMP	2	4.2% (48)
		HOMDEL	1	2.1% (48)
Pediatric Acute Lymphoid Leukemia	*CREBBP*	AMP	2	0.3% (764)
		HOMDEL	8	1.0% (764)
	*EP300*	AMP	3	0.4% (764)
		HOMDEL	3	0.4% (764)
Pediatric Acute Myeloid Leukemia	*CREBBP*	AMP	1	0.4% (240)
	*EP300*	HOMDEL	2	0.8% (240)
Tumor	Gene	Structural variant number	Case number with mutation	Percentage (total number)
Acute Lymphoblastic Leukemia	*CREBBP*	1	1	1.1% (93)
Acute Myeloid Leukemia	*CREBBP*	1	1	0.5% (200)

Notes: Cytoband: *CREBBP* (16p13.3); *EP300* (22q13.2); CNA: copy number alteration; HOMDEL: homozygous deletion; AMP: amplification. The above data mainly come from the TCGA database.

**Table 3 cancers-15-01219-t003:** CREBBP/EP300 and oncogenic protein complexes in leukemia and mechanisms.

Complex Components	Tumors	Mechanisms	References
ZNF384-CREBBP t(12;16) (p13;p13) and ZNF384-EP300 t(12;22)	ALL	Upregulating JAK/STAT and cell adhesion pathways, downregulating cell cycle and DNA repair pathways, and enhancing oncogenic transformation	[81,82]
TCF3-HLF	ALL	Preferentially cooperating with ERG to recruit EP300 to activate the gene expression critical to ALL, which is also associated with chemoresistance	[83]
MAFB-ETS2	T-ALL	Interacting with PCAF and EP300 to enhance NOTCH1 signaling, including *MYC*, *NOTCH3*, and *HES1*	[84]
PML-RARα	APL	Recruiting abundant EP300 and HDAC1 to target genes, such as *GFI1*, exerts an activating effect by forming super-enhancers, while only sufficient HDAC1 is recruited to repressed target genes, such as *CEBPE*, which will exhibit repressive effects	[85]
MYB-C/EBPβ-EP300	AML	*GFI1*, a target gene of MYB-C/EBPβ-EP300, is downregulated by C/EBPβ-inhibitory natural sesquiterpene lactones, further inhibiting cell proliferation	[86]
RUNX1-ETV6	ALL	Inducing leukemogenesis through acetylation of RUNX1 by mTORC1 phosphorylated EP300	[87]
RUNX1-ETO	AML	EP300 colocalizes in the regulatory regions of many RUNX1-ETO target genes and acetylates RUNX1-ETO to promote leukemogenesis	[88]
		Recruiting EP300 to activate *THAP10*, which is a target and negatively regulated by *microRNA-383*	[89]
E2A-PBX1	ALL	Recruiting EP300, H3K27ac, and MED1 to E2A-PBX1-targeted RUNX1 sites	[90]
MOZ–CREBBP t(8;16)(p11;p13)	AML	Causing upregulation of *HOXA* family, *PBX3*, *MEIS1*, *HNMT*, etc., and inhibiting RUNX1-mediated differentiation of M1 myeloid cells into monocytes/macrophages	[79,80]
MOZ-TIF2 and Nup98-Hoxa9	AML	Recruiting CREBBP/EP300 to induce leukemogenesis and serving as therapeutic targets for various human AML subtypes	[91]
MOZ-TIF2 and MLL-AFX	AML	Recruiting the AF4 family/ENL family/P-TEFb complex and activating CpG-rich promoters by CREBBP/EP300	[92]

Notes: ALL: acute lymphocytic leukemia; T-ALL: T-cell acute lymphocytic leukemia; APL: acute promyelocytic leukemia; AML: acute myeloid leukemia.

**Table 4 cancers-15-01219-t004:** Inhibitors of CREBBP/EP300 in hematological malignancies and their functions.

Inhibitors	Targets	Tumor or Cells	Functions	Reference
CCS1477, CU329	BRD/HAT	DLBCL cells	Reducing CREBBP/EP300 autoacetylation, H3K18ac, and H3K27ac, and significant negative enrichment in CREBBP- and EP300-regulated programs	[49]
CPI-637	BRD	ALCL and HL	Inhibiting PD-L1 mediated tumor immune escape in vitro and in vivo	[112]
GNE272	BRD	lymphoma and MM cells	Repressing expression of multiple oncogenes, including *MYC*, *MYB*, *CCND1*, *BCL2*, *BCL-XL*, *MCL1*, and *BCL6*, and exerting a significant antiproliferative effect on some tumor cells, such as Jurkat, Pfeiffer, KMS11, and U266B1	[113]
GNE-781	BRD	FL	Reducing differentiation of human CD4^+^ T-cells into Tregs, impairing proliferation of both naive activated T-cells and induced Tregs, and reducing cytokine secretion, such as prostacyclin, IL10, and IL2	[114]
NEO2734, NEO1132	BRD	MM	Effectively decreasing both cMyc and IRF4 protein expression and inducing G1 cell cycle arrest	[115]
SGC-CBP30	BRD	AML	Reducing binding of BRD4 to the regulatory region of B7-H6 and decreasing expression of B7-H6	[116]
SGC-CBP30	BRD	MM	Inhibiting IL6 autocrine production and STAT3 activation, downregulating IRF4 (especially truncated IRF4) and MYC, and restoring immunomodulatory drugs sensitivity	[101]
SGC-CBP30, I-CBP112	BRD	MM cells	Upregulating cell surface and mRNA expression of *MICA* and exhibiting an anti-myeloma effect by inhibiting the IRF4/MYC axis	[99,117,118]
SGC-CBP30, GNE-049	BRD	AML cells	Blocking H3K27ac, eRNA production, and expression of enhancer-proximal genes	[13]
SGC-CBP30, GNE-272, CPI644	BRD	CML cells	Inhibiting the expression of super-enhancer-associated genes such as *MYC*, *GATA1*, and *MYB*, as well as downregulating the expression of their target genes, such as *TET1*, *FOSL1*, and *CCND1*	[5]
XDM-CBP	BRD	leukemia	Inhibiting cancer cells proliferation	[119]
A485	HAT	ALL, MM, AML and B-NHL	Blocking TCF3-HLF-dependent gene expression, including an *MYC*-associated signature, and inhibiting tumor cell proliferation	[83,120]
A-485, A-241	HAT	CLL and MM cells	Inducing potent apoptotic and cytostatic effects, suppressing IRF4-dependent MM signatures such as MYC, and PRDM1/BLIMP-1, and downregulating gene expression including those regulating B-cell activation and known oncogenic drivers in CLL, such as *IRC3*, *ID3*, and *MYC*	[54]
A485, C646	HAT	AML	Relieving myeloid differentiation abnormalities associated with HMGN1 overexpression in hematopoietic progenitor cells and leukemia	[93]
B026	HAT	Leukemia and lymphoma cells	Robustly decreasing *MYC* expression and growth of cell lines such as MV-4-11, Maver-1, K562, and Kasumi-1	[121]
C646	HAT	ALL	Attenuating *GNAO1* expression upregulated by ETV6-RUNX1	[87]
C646	HAT	AML	Suppressing growth and colony formation in multiple AML cell lines and primary human AML samples	[91]
C646	HAT	DLBCL cells	Inhibiting H3K9ac and H3K14ac and RUNX1-ETO while increasing THAP10 levels	[89]
C646	HAT	MDS-derived AML cells	Increasing the sensitivity of AML cells to azacitidine	[122]
Salicylate, diflunisal	HAT	DLBCL	Inhibiting CREBBP/EP300 lysine acetyltransferase activity by direct competition with acyl-CoA at the catalytic site and suppressing the growth of EP300-dependent leukemia cell lines expressing RUNX1-ETO fusion protein in vitro and in vivo	[123]

Notes: BRD: bromodomain; HAT: histone acetyltransferase; ALCL: anaplastic large cell lymphoma; HL: Hodgkin lymphoma; DLBCL: diffuse large B-cell lymphoma; FL: follicular lymphoma; MM: multiple myeloma; CML: chronic myeloid leukemia; B-NHL: B-cell non-Hodgkin lymphoma; CLL: chronic lymphocytic leukemia; MDS: myelodysplastic syndrome.

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
