# Peer review of "The Role of CREBBP/EP300 and Its Therapeutic Implications in Hematological Malignancies"

_cancers, 2023, doi:10.3390/cancers15041219_

Round 1

Reviewer 1 Report

 It has been summarized how different inhibitors of CREBBP/EP300 and histone deacetylase can target therapeutic potential, alleviate chemotherapy resistance, and enhance immunotherapeutic potential.

-The names of CREBBP/EP300 inhibitors should also be included in the manuscript and brief information should be given.

-On the basis of this review article, the authors need to specify a clear research space or gap that still requires further research.

-The meaning of some abbreviations is not included in the manuscript such as MHCI.

Author Response

回复审核人#1

评论1:

CREBBP/EP300 防腐剂的名称也应包含在手稿中并提供简单的信息。

响应1:

感谢您改进手稿的重要重要建议建议。。。。。。。了了了了了了了了了了等综述综述综述综述综述综述,crebbp/ep300的的/ep300抑制剂这综述中中中中中中中,crebbp/ep300 brd hat抑制剂和和和和和和抑制剂抑制剂抑制剂信息。

评论2:

在这篇综述文章的基础上,作者需要明确一个尚待一步研究的研究空间或差距。

响应2:

感谢对提出的建设性意见意见建议建议建议建议,我们建议建议建议建议建议建议建议建议当前当前当前手稿手稿手稿中中中中中提出提出提出提出了了以下(1) (2) crebbp /ep300在在在在在在在在转录组转录组水平广泛的的的造血调控网络网络这两两种种酶酶酶酶酶(2)有待揭示(4.2节第一段)。(3) JQAD1和dCBP-1(CREBBP/EP300的两种化学降解剂)在血液系统恶性肿瘤中的疗效和机制需要一步一步探索()。(4)crebbp/ep300抑制剂逐渐增多增多和增多增多增多(5)由于。少少少少少,需要研究研究研究(((。。(5)由于由于由于由于。。(5)由于由于高高高高可可变性变性变性变性变性变性变性变性。。的数量数量少少少少少少少少少少少少, crebbp/ep300基因组基于高高高高可能会增强 CREBBP/EP300 的治疗潜力。

评论3:

一些缩写词的含义在手稿中没有包,例如MHCI。

响应3:

感谢您的详细审查。我们为文本中缺失的缩写添加了确定性,包括MHCI和MHCII。

Reviewer 2 Report

this is a well-written and informative manuscript that summarizes the role of CREBBP/EP300 in pathogenesis and drug resistance in various leukemia conditions. 

Author Response

Reply to Reviewer #2

Comments

This is a well-written and informative manuscript that summarizes the role of CREBBP/EP300 in pathogenesis and drug resistance in various leukemia conditions.

Response

We sincerely thank you for your review of our manuscript and your valuable comments. We have checked and corrected the language problems in the manuscript.

Reviewer 3 Report

The review article titled “The role of CREBBP/EP300 and its therapeutic implications in hematological malignancies” by Zhu et al. is all about, the characterization of CREBBP/EP300, evolutionarily conserved and functionally related enzymes. A comprehensive overview of how CREBBP/EP300 contributes to the genesis and progression of hematologic malignancies. Also, the authors discussed the CREBBP/EP300 inhibitors and histone deacetylase inhibitors on targeting therapeutic potential, alleviating chemotherapy resistance, and enhancing immunotherapeutic potential has also been reviewed. The review article is well-written and discussed the CREBBP/EP300 therapeutic modalities in hematological malignancies. The point-wise comments are as follows.

1.       CREBBP/EP300 serve as tumor activators or suppressors depending upon the situation. In what situation do these enzymes works as activators or suppressor?

2.       Most of the therapeutic modalities are working as inhibitors, which means it is working as activators. Is there any example/case where these two enzymes worked as suppressors and any activator worked as a therapeutic option?

3.       CREBBP/EP300 is an enzyme that playing role in hematological malignancies. What is the baseline expression level of these two enzymes in the bloodstream?

4.        Is there any evidence of modulation of expression of these two enzymes playing role in hematological malignancies? 

Author Response

Reply to Reviewer #3

Comment 1:

CREBBP/EP300 serve as tumor activators or suppressors depending upon the situation. In what situation do these enzymes works as activators or suppressor?

Response 1:

Thank you for your great suggestions to improve our manuscript. In the revised manuscript, we show that abnormal expressions of CREBBP/EP300 are always associated with tumorigenesis and progression, acting as tumor activators. In your second comment, we listed several examples of CREBBP/EP300 working as tumor suppressors.

Comment 2:

Most of the therapeutic modalities are working as inhibitors, which means it is working as activators. Is there any example/case where these two enzymes worked as suppressors and any activator worked as a therapeutic option?

Response 2:

Thank you for your great suggestions to improve the content of our manuscript. Based on your comments, we have given the following examples in the manuscript. (1) EP300 also has tumor suppressor effects in several different clinically relevant MDS models driven by mutations in epigenetic regulators TET2, ASXL1, and SRSF2, respectively, inhibiting the malignant transition of MDS to AML. This suggests a potential therapeutic application of EP300 agonists in the treatment of MDS with the aforementioned inactivating mutations (3.4. Myelodysplastic Syndromes). (2) EP300, but not CREBBP, plays a critical role in blocking the transformation of myelodysplastic syndrome (MDS) to AML and in controlling the balance between symmetric stem cell self-renewing divisions and stem cell depleting divisions in Nup98-HoxD13 transgenic mice, an animal model that phenotypically replicates human MDS (Figure 3C and Paragraph 3 of Section 3.2). (3) Aberrant expression of CREBBP/EP300 causes Rubinstein–Taybi syndrome and multiple hematologic malignancies by losing their activity. CREBBP is a haploinsufficient tumor suppressor gene in GC B-cells. CREBBP deficiency promotes the development of B-cell lymphoma. EP300 could suppress the transition of MDS to AML, suggesting the therapeutic potential of EP300 agonists in MDS patients with Tet2 inactivated mutations. YF-2 is a highly selective CREBBP/EP300 HAT domain agonist, and it could improve the genome editing efficiency of cas9 nucleases. CTB, an agonist of EP300, increases the expression of EP300 to induce the acetylation of P53 protein and subsequently leads to the death of breast cancer cells. Therefore, CTB can be used as an anticancer drug. TTK21 is also a CREBBP/EP300 agonist that conjugates glucose-based carbon nanosphere (CSP) to cross the blood-brain barrier, which is beneficial for adult neurogenesis and long-term memory function. Currently, the number of CREBBP/EP300 agonists is relatively rare compared to inhibitors, and more in-depth studies are required.

Comment 3:

CREBBP/EP300 is an enzyme that playing role in hematological malignancies. What is the baseline expression level of these two enzymes in the bloodstream?

Response 3:

We thank you for the critical comment and helpful suggestion for our manuscript. By searching relevant journals and databases, we found that CERBBP/EP300 were two enzymes with low tissue specificity. They are widely expressed in and outside the blood system and are moderately expressed in bone marrow and lymphatic tissues compared to other tissues. We have described in the introduction section that CREBBP/EP300 are widely expressed both within and outside the blood system.

Comment 4:

Is there any evidence of modulation of expression of these two enzymes playing role in hematological malignancies?

Response 4:

We are grateful for this suggestion. In the revised manuscript version, we can find the following evidence that modulation of expression of these two enzymes plays a role in hematologic malignancies. (1) Genes disturbed by CREBBP mutation are direct targets of the BCL6/SMRT/HDAC3 tumor-repressor complex. CREBBP loss of function contributes to lymphomagenesis by promoting immune escape in vitro and in vivo (Figure 2 and Paragraph 2 of Section 3.1). (2) Aberrant expression of CREBBP/EP300 causes Rubinstein–Taybi syndrome and multiple hematologic malignancies by losing their activity. CREBBP is a haploinsufficient tumor suppressor gene in GC B-cells. CREBBP deficiency promotes the development of B-cell lymphoma (Paragraph 1 of Section 4.1.1). (3) Downregulating CREBBP inhibits ALL cell proliferation and cell cycle progression and leads to daunorubicin resistance by interacting with E2F3a (Paragraph 1 of Section 4.2).

Reviewer 4 Report

This is a comprehensive review of the role of CREBBP/EP300 in hematological malignancies, including possibilities for treatment. 

Author Response

Reply to Reviewer #4

Comments

This is a comprehensive review of the role of CREBBP/EP300 in hematological malignancies, including possibilities for treatment.

Response

We feel great thanks for your professional review work on our manuscript. In the revised version, we have corrected the English language problems in the manuscript.

Round 2

Reviewer 1 Report

it is acceptable in its present form.